# High-performance thermomagnetic generator controlled by a magnetocaloric switch

Xianliang Liu[1], Haodong Chen[1], Jianyi Huang[1], Kaiming Qiao[1], Ziyuan Yu[1], Longlong Xie[1], Raju V. Ramanujan[2], Fengxia Hu [3,4,5], Ke Chu[6], Yi Long[1] & Hu Zhang [1] ✉

Low grade waste heat accounts for ~65% of total waste heat, but conventional waste heat recovery technology exhibits low conversion efficiency for low grade waste heat recovery. Hence, we designed a thermomagnetic generator for such applications. Unlike its usual role as the coil core or big magnetic yoke in previous works, here the magnetocaloric material acts as a switch that controls the magnetic circuit. This makes it not only have the advantage of flux reversal of the pretzel-like topology, but also present a simpler design, lower magnetic stray field, and higher performance by using less magnetocaloric material than preceding devices. The effects of key structural and system parameters were studied through a combination of experiments and finite element simulations. The optimized max power density $P_{Dmax}$ produced by our device is significantly higher than those of other existing active thermomagnetic, thermo, and pyroelectric generators. Such high performance shows the effectiveness of our topology design of magnetic circuit with magnetocaloric switch.

The globe has invested a lot of effort, money, and other resources to reduce emissions in order to alleviate the greenhouse effect. However, waste heat is a major source for recycled energy, offering great potential for reducing greenhouse gas emissions[1,2]. Recent research indicates that 49.3-51.5% of global energy consumption will eventually become waste heat. Usually, high temperature waste heat is easy to be recovered due to its high energy level, while low-temperature waste heat is hard to be re-utilized because of the low energy level and few efficient recovery technologies. However, it should be noted that the low-temperature waste heat, with a temperature below 100 °C, accounts for ~65% of the total waste heat[3]. Thus, it is necessary to search for low-grade waste heat recovery technology.

In comparison with the direct use of waste heat, the conversion of heat into electricity not only avoids the heat dissipation during transmission, but also facilitates the energy storage. Unfortunately, few

technologies can efficiently convert low-grade waste heat into electricity. Recently, the pyroelectric generator (PEG) has attracted much attention as a heat energy recovery technology based on the charge release produced by the polarization change of pyroelectric materials with temperature. Although PEG has the advantage of high-frequency response, its low output current and high impedance limit its practical applications[4,5]. In addition, the thermoelectric generator (TEG) based on the Seebeck effect has been considered as another promising waste heat recovery technology. But its conversion efficiency reduces significantly with the decrease of temperature difference, which hinders its application at low-grade waste heat temperatures[6–8].

In the 19th century, Edison[9] and Tesla[10] proposed the concept of the thermomagnetic generator (TMG), which could convert heat into electricity based on the Faraday's law of electromagnetic induction. In principle, the TMG performance mainly relies on the temperature-

[1]School of Materials Science and Engineering, University of Science and Technology Beijing, Beijing 100083, P R China. [2]School of Materials Science and Engineering, Nanyang Technological University, Singapore 639798, Singapore. [3]Beijing National Laboratory for Condensed Matter Physics, Institute of Physics, Chinese Academy of Sciences, Beijing 100190, P R China. [4]School of Physical Sciences, University of Chinese Academy of Sciences, Beijing 100049, P R China. [5]Songshan Lake Materials Laboratory, Dongguan, Guangdong 523808, P R China. [6]School of Materials Science and Engineering, Lanzhou Jiaotong University, Lanzhou 730070, P R China. ✉e-mail: zhanghu@ustb.edu.cn

driven phase transition of the thermomagnetic material (TMM) and the associated large change in magnetization around the Curie temperature ($T_C$)[11,12]. Theoretical studies show that the conversion efficiency of TMG could reach as high as 55% of the Carnot efficiency[13]. Therefore, it is more suitable for low-grade waste heat recovery. However, in the past decades, the practical development of TMG was very slow due to the lack of suitable TMMs with Curie temperatures near room temperature. Besides, the factors such as the design of the magnetic circuit, optimization of material and device parameters, and actual operating conditions result in the challenges in transferring theory to practice.

TMGs can be divided into passive TMG and active TMG. The difference between the two is that TMG converts thermal energy into electricity either directly (active devices) or indirectly (passive devices) via mechanical motion[14]. The passive TMG realizes self-driven vibration under the combined action of the temperature gradient, magnetic force, and the elastic force of the cantilever by combining the TMM with the cantilever beam. The TMG can be combined with a piezoelectric material (PEM) or triboelectric material (TEM) to convert mechanical vibration into electrical energy[11,15,16]. Nevertheless, the output power of the passive TMG is low due to the high impedance of PEM and TEM as well as due to the energy loss during indirect conversion[11,16]. On the contrary, active TMG exhibits higher theoretical conversion efficiency than passive TMG since it utilizes the magnetization change of TMM to directly convert thermal energy into electrical energy[12-14,17-19]. Unfortunately, the practical performance of the previous active TMGs is low due to the limited magnetic flux change in the magnetic circuit[12,17,19]. The optimization of the magnetic circuit structure was lacking. In our recent work, we studied the effects of device and material parameters on the traditional active TMG performance, and improved the TMG performance significantly by optimizing key parameters[20]. Different from traditional TMG, Waske et al. constructed a pretzel-like magnetic circuit to reverse the flow direction of the magnetic flux in the magnetic circuit. Compared with the traditional magnetic circuit device, the output power is improved by an order of magnitude. However, the maximum relative Carnot efficiency of TMG only reaches $1.7 \times 10^{-3}$%, which is still a huge gap compared to the theoretical value of 55%[18]. It suggests that the performance of TMG still has great optimization potential.

Here, we designed a TMG for low-temperature waste heat recovery. The TMG design utilizes magnetocaloric material as a small magnetic circuit switch. This design not only achieves the reversal of magnetic flux like the pretzel-like topology[18], but also shows a simpler design and lower magnetic stray field. In this context, we use finite element simulation to investigate the effect of key structural parameters on the magnetic circuit performance, and then fabricated a TMG device based on the optimized parameters. The experimental results demonstrate that this TMG exhibits a much higher performance than previous TMGs;[11,12,16,17,21,22] the max power density is 2 to 3 orders of magnitude higher than those of previous reports[12,17,21]. It also exhibits higher power density in comparison with TEG[23-26] and PEG[5,27,28] technologies. Finally, the electricity generated by this TMG successfully lights up a LED, which strongly proves its practicality.

## Results and Discussion
### Topology design of magnetic circuit with MCS
Figure 1a shows the topology of the magnetic circuit in the present TMG. The magnetic circuit consists of three yokes, which are connected by small permanent magnets and layered TMMs. The permanent magnets with a remanence of -1.2 T provide a stable magnetic field. The magnetic flux circulates through a complete closed loop, which lowers magnetic stray fields. The typical magnetocaloric material Gd is chosen as the TMM, and it undergoes a second-order magnetic transition between the ferromagnetic (FM) and the paramagnetic (PM) states around $T_C$ = 292 K (Supplementary Fig. 1). Instead of acting as the coil core as in previous TMGs[12,17,19], the Gd is used to connect the permanent magnet and the central yoke, which is wound by an induction coil. The Gd acts as a switch, controlling the ON/OFF of the magnetic circuit, so we denote it the magnetocaloric switch (MCS). In our design, we do not need a large amount of magnetocaloric material to improve the magnetic flux change of the coil. The soft magnetic iron core is cheap and easy to process and has a higher saturation magnetization of 2.2 T compared to that of Gd (-1.4 T, Supplementary Fig. 1), which is favorable to obtain a large magnetic flux change in the coil.

In order to facilitate the heat exchange, the MCS is made into a layered structure with Gd thin plates. Hot and cold water are controlled to flow alternately through the right and left MCSs, thus switching the circuit on each side. As shown in Fig. 1b, when cold

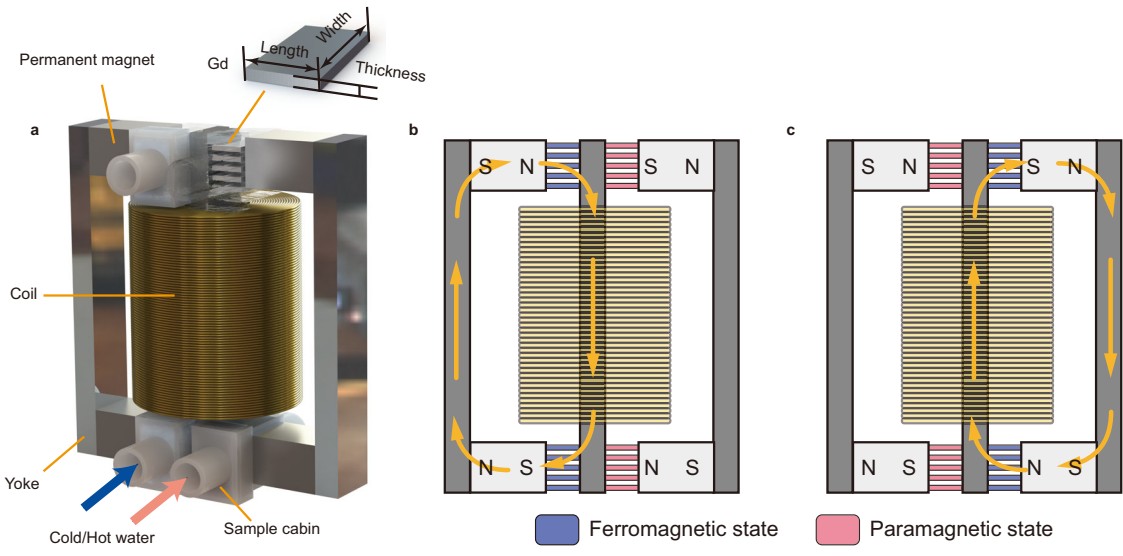

**Fig. 1 | Magnetic circuit structure design and working principle of the TMG. a** Detailed schematic illustration. the main components of the magnetic circuit include the NdFeB permanent magnets, magnetic yokes, gadolinium (Gd), sample cabins, and a coil. **b, c** The working principle of the magnetic circuit during the cycle. Blue indicates that the TMM is ferromagnetic at low temperature, and red indicates that the TMM is paramagnetic at high temperature. Since the temperature of the TMMs on both sides alternates between cold and hot, the direction of the magnetic flux in the magnetic circuit is reversed.

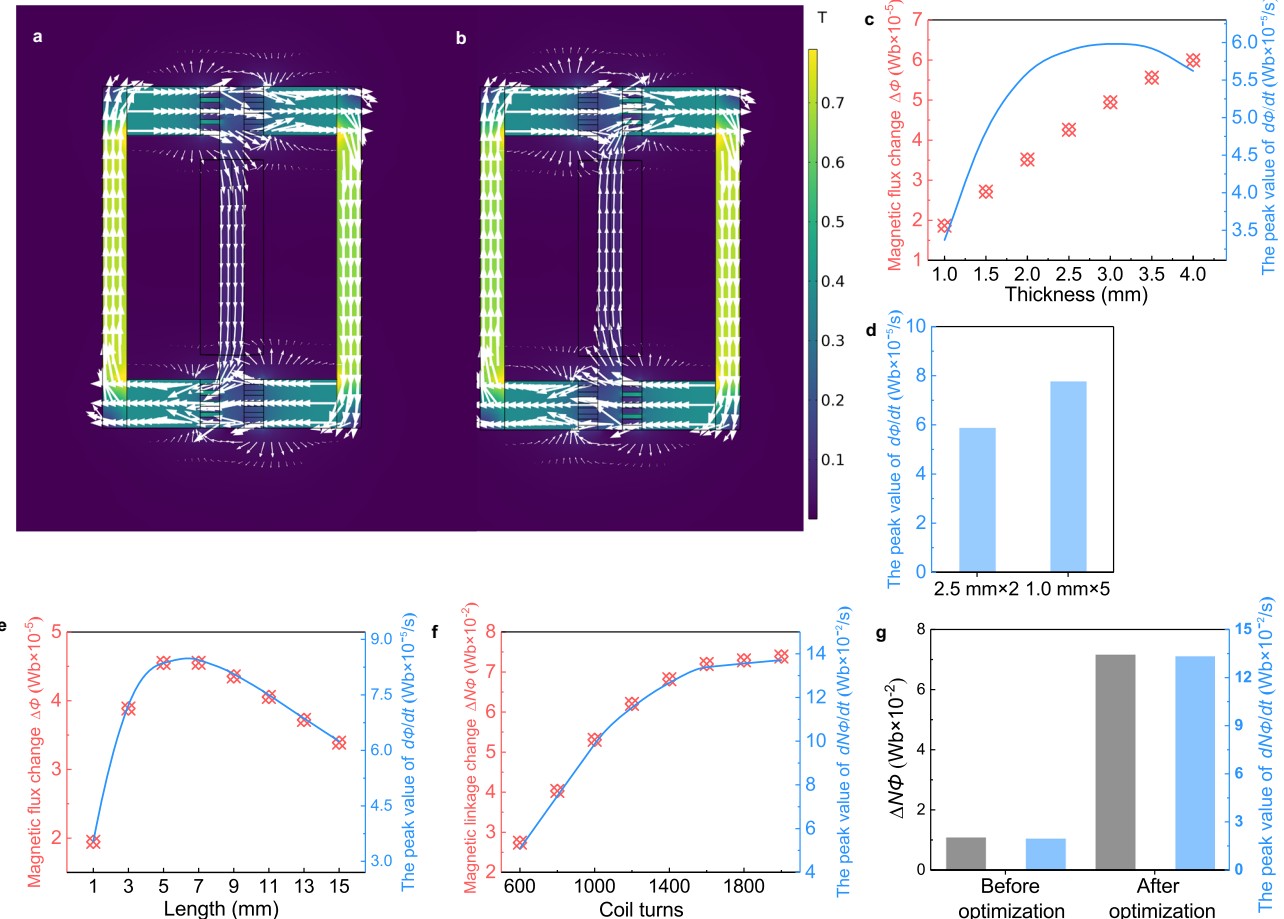

**Fig. 2 | Simulation results of the magnetic circuit in the TMG. a, b** The distribution of magnetic flux density **B** in the magnetic circuit under two different states. The white arrow represents the direction of magnetic flux in the magnetic circuit. Whenever the temperature of the TMMs on either side alternates between hot and cold, the direction of magnetic flux in the magnetic circuit is reversed. **c** The effect of the thickness on magnetic circuit performance. When the magnetic flux density **B** is constant, the larger the thickness of the TMM, the larger the conducted magnetic flux $\Phi$. **d** The peak value of the $d\Phi/dt$ of two 2.5 mm thick samples in each sample cabin and five 1 mm thick samples in each sample cabin. **e** The effect of the length of the TMM on the magnetic circuit. **f** The effect of coil turns on the magnetic circuit. **g** Comparison of magnetic circuit performance before and after structural optimization. (Source data are provided as a Source Data file).

water flows through the left circuit while hot water flows through the right circuit, the left circuit is switched ON while the right one is turned OFF due to the magnetic transition of Gd. In this case, the magnetic flux flows clockwise in the left circuit. When the hot and cold water are switched, as shown in Fig. 1c, the left circuit is switched OFF and the right one is turned ON. The magnetic flux flows in the right circuit. This topology makes the magnetic flux in the induction coil wound around the center yoke change between the negative and positive maximum value, like the effect of the pretzel-like topology[18]. In comparison with most previous TMGs in which the flux only changes between zero and maximum value, the present topology increased the induced power by a factor of four. This topology has all the advantages of the pretzel-like topology. Moreover, it shows more advantages than pretzel-like topology. For example, (1) a single coil in the core yoke is a simpler design than the complicated pretzel-like design with multiple coils around different yokes; (2) unlike acting as the big magnetic yoke in pretzel-like topology device, much fewer materials are needed to produce the small MCS in our design, which lowers the preparation cost and difficulty; and moreover, (3) small MCSs also avoids greater magnetic stray when magnetic flux pass through the big stacked magnetocaloric yoke, and then increases the induced voltage. In addition, unlike pretzel-like topology using only half of the flux change in the closed magnetic circuit, all the magnetic flux in the

closed circuit is utilized in our TMG. These advantages make the present design exhibit a higher performance than previously reported TMGs.

## Optimization of structural parameters

The induced electrodynamic potential ($V$) can be determined by the following equation according to the Faraday's law[12,29],

$$V = -N\frac{d\Phi}{dt} \tag{1}$$

where $N$ is the turns of the coil and $d\Phi/dt$ is the rate of change of the magnetic flux. The $d\Phi/dt$ is related to the design of the magnetic flux topology and strongly affected by the structural parameters, such as the thickness and length of the TMM. Therefore, we constructed the magnetic circuit structure of the TMG using finite element simulations and studied the effect of the structural parameters on the TMG performance in order to select the optimal parameters. The geometry, boundary conditions (Supplementary Fig. 2), and mesh strategy (Supplementary Fig. 3) of the numerical model are detailed in Supplementary Note 2. The magnetic flux density distribution images during heating and cooling were obtained by the simulation as displayed in Fig. 2a,b. The magnetic flux flows clockwise in the left circuit while the right circuit is not conductive at the TMM position when cold

water flows through the left circuit while hot water flows through the right circuit. On the other hand, the left circuit is switched OFF while the right one is turned ON when the hot and cold water are reversed. The magnetic flux in the center yoke is reversed when the hot and cold water are alternated. In comparison with the traditional magnetic circuit design shown in Supplementary Fig. 4, the magnetic flux change $\Delta\Phi$ in the present topology is twice of the traditional magnetic circuit.

Two Gd plates with length × width × thickness of $10 \times 12 \times 1$ mm were used as MCSs in the initial simulation. Figure 2c and Supplementary Fig. 5 show that as the thickness of the MCS increases from 1 mm to 4 mm, the magnetic flux change $\Delta\Phi$ in the magnetic circuit increases linearly from $1.86 \times 10^{-5}$ Wb to $6.0 \times 10^{-5}$ Wb. However, the peak value of the magnetic flux change rate $d\Phi/dt$ increases first with the thickness, and then gradually decreases after reaching a maximum of $6.0 \times 10^{-5}$ Wb/s at a thickness of 3.0 mm. Although larger thickness would enhance the $\Delta\Phi$, a thicker TMM plate would also lower the rate of temperature change $dT/dt$. As shown in Supplementary Fig. 5a, b, $dT/dt$ of the MCS decreases distinctly for greater thickness, which is opposite to the change of $\Delta\Phi$. Considering that $\frac{d\Phi}{dt} = \frac{d\Phi}{dT} \cdot \frac{dT}{dt}$, larger $d\Phi/dT$ with the increase of thickness results in the increase of $d\Phi/dt$ at first, and then the decrease of $dT/dt$ becomes more predominant with larger thickness, leading to a maximum followed by a gradual decrease of $d\Phi/dt$. This result suggests that, when increasing the volume of TMM to increase the $\Delta\Phi$, it is also necessary to consider using thin plates to ensure a large specific surface area, which could promote the heat exchange and obtain a large $dT/dt$. In order to prove it, we compared the performance of two 2.5 mm thick samples in each sample cabin and five 1 mm thick samples in each sample cabin, as shown in Fig. 2d and Supplementary Fig. 6. In this case of the same cross-sectional area, the magnetic flux change $\Delta\Phi$ in the magnetic circuit remains unchanged, while the temperature change rate $dT/dt$ increases with smaller thickness of the MCS plate, resulting in a higher magnetic flux change rate $d\Phi/dt$. The peak value of $d\Phi/dt$ increases from $5.9 \times 10^{-5}$ to $7.8 \times 10^{-5}$ Wb/s. This result confirms that layered thin plates are favorable to higher TMG performance.

By keeping the five 1 mm thick plates in the sample cabin, the effect of the MCS length on the $d\Phi/dt$ is further studied by simulation as shown in Fig. 2e and Supplementary Fig. 7. The magnetic flux change $\Delta\Phi$ increases from $1.94 \times 10^{-5}$ Wb for a length of 1 mm to a peak value of ~$4.55 \times 10^{-5}$ Wb in the range of $5 \sim 7$ mm. With a further increase in length, the magnetic flux change $\Delta\Phi$ starts to decrease gradually. The magnetic flux density distributions diagram for different length of MCS are displayed in Supplementary Fig. 8 in order to understand the variation of $\Delta\Phi$. When the length of MCS is 1 mm, such a short distance between the permanent magnet and the center yoke could cause a large amount of flux to flow through the right MCS even if it is in the PM state, i.e., the magnetic flux densities **B** of the left and right MCSs are 0.4 T (ferromagnetic state) and 0.26 T (paramagnetic state), respectively. A large value of $\Phi$ of $5.9 \times 10^{-5}$ Wb circulates along the outer loop, while only a small $\Phi$ of $0.97 \times 10^{-5}$ Wb flows through the center yoke. With an increase of length, it is harder to magnetize the right MCS at PM state, more magnetic flux is shunted from the outer loop to the center yoke, resulting in a distinct increase of $\Phi$ through the center yoke. It is seen from Supplementary Fig. 8 that almost no flux flows through the right MCS when the length reaches 7 mm, indicating that the MCS is completely OFF. Thus, the $\Delta\Phi$ reaches the maximum value of ~$4.55 \times 10^{-5}$ Wb. However, when the length is larger than 7 mm, the permanent magnet cannot fully magnetize the MCS, and a large amount of magnetic flux is strayed into the air. The magnetic flux in the magnetic circuit decreases gradually. Since the thickness remains at 1 mm, heat transfer from the surface to the core will not change, the $dT/dt$ of the MCS remains unchanged. The $d\Phi/dt$ value is only affected by the magnetic flux change $\Delta\Phi$, thus the variation of $d\Phi/dt$ is consistent with that of $\Delta\Phi$, as shown in Fig. 2e.

In addition to $d\Phi/dt$, it has pointed out in ref. 13 that the coil needs to meet the condition $R = \omega L$ for optimal power output, where $R$ is the resistance in the circuit, $\omega$ is the operating frequency, and $L$ is the inductance of the coil. When only the coil resistance is considered, the above formula can be converted into $N = \rho\pi(D_1 + D_2)h/2A\omega\mu_a S$, where $\rho$ is the specific resistance of copper, $D_1$ and $D_2$ are the inner and external diameter of the coil, $h$ is the length of the coil, $A$ is the cross-section area of the wire, $\mu_a$ is the permeability of the coil, and $S$ is the cross-section area of the coil. In another word, the number of coil turns $N$ should meet above condition for optimal power output, and so it is an important parameter for optimizing the power output. The effect of coil turns $N$ on the TMG performance is investigated as shown in Fig. 2f and Supplementary Fig. 9. The flux linkage change $\Delta N\Phi$ (the magnetic flux in the coil multiplied by the number of turns) and the peak value of the magnetic linkage change rate $dN\Phi/dt$ increase almost linearly as $N$ increases from 600 to 1400. Then, both $\Delta N\Phi$ and $dN\Phi/dt$ tend to saturate with further increase in $N$ higher than 1600, and reach values of $7.38 \times 10^{-2}$ Wb and $13.38 \times 10^{-2}$ Wb/s, respectively. The surface coil is about ~2 cm away from the center yoke when $N > 1600$, so these coils are unable effectively sense flux changes in the center yoke (Supplementary Fig. 9a), resulting in saturation of $\Delta N\Phi$ and $dN\Phi/dt$.

Based on above analysis, we summarize the effects of key parameters on the $dN\Phi/dt$ and then choose the optimized parameters (Supplementary Fig. 10).

a. An increase of thickness would increase the $\Delta\Phi$, but the $dT/dt$ would decrease with the larger thickness, lowering the $dN\Phi/dt$. So, 1 mm × 5 layered plates are chosen as the MCS.

b. We choose 6.5 mm as the length of the MCS to avoid the shunt magnetic flux in the outer loop as well as the magnetic stray fields.

c. The coil turns $N$ is the most significant factor on the TMG performance, e.g., the $dN\Phi/dt$ increases by 83.1% with a 100% increase of $N$. But the surface coil cannot sense flux changes in the center yoke when $N > 1600$. Therefore, a coil turn of 1600 is chosen as the optimized parameter.

The parameters before and after optimization are listed in Supplementary Table 1. The $\Delta N\Phi$ and $dN\Phi/dt$ before and after optimization are compared as shown in Fig. 2g. The $\Delta N\Phi$ increases from $1.12 \times 10^{-2}$ Wb to $7.19 \times 10^{-2}$ Wb, and the peak value of $dN\Phi/dt$ increases from $2.02 \times 10^{-2}$ Wb/s to $13.38 \times 10^{-2}$ Wb/s, indicating that the induced $V$ would increase by 562 %, according to Eq. (1).

## Construction and characterization of TMG

We built the TMG magnetic circuit using the optimized structural parameters and assembled the auxiliary system to test the TMG performance. As shown in Fig. 3a–d, 5 Gd plates of $6.5 \times 12 \times 1$ mm³ are arranged in layers in each sample cabin, which is 3D printed with nylon. The center yoke is wound by a copper coil of 1600 turns. Hot and cold water are pumped to flow through the left and right MCSs at the same time (Fig. 3e), to turn the left MCS OFF and the right MCS ON by the thermal-induced magnetic transition. Then, the hot and cold water are switched by the controlled valves to turn the left MCS ON and the right MCS OFF. The magnetic flux in the center yoke is reversed, resulting in a large induced $V$ in the coil. It is noted in Fig. 3e–g that we define the lower side as the left side, the upper side is the right side. As shown by the real-time infrared image in Fig. 3f, the hot and cold-water temperatures at the right and left inlets are ~84.4 °C (357.4 K) and -17 °C (290 K), respectively. The temperatures distribution is switched after switching the hot and cold water (Fig. 3g).

To measure the temperature in the sample cabin, thermocouples were used to measure both water and sample temperatures in the cabin. Figure 3h,i show the water and sample temperatures on the left and right sides, respectively. The sample temperature is consistent with the water temperature, revealing good thermal conduction. The temperatures on the left and right sides change simultaneously in

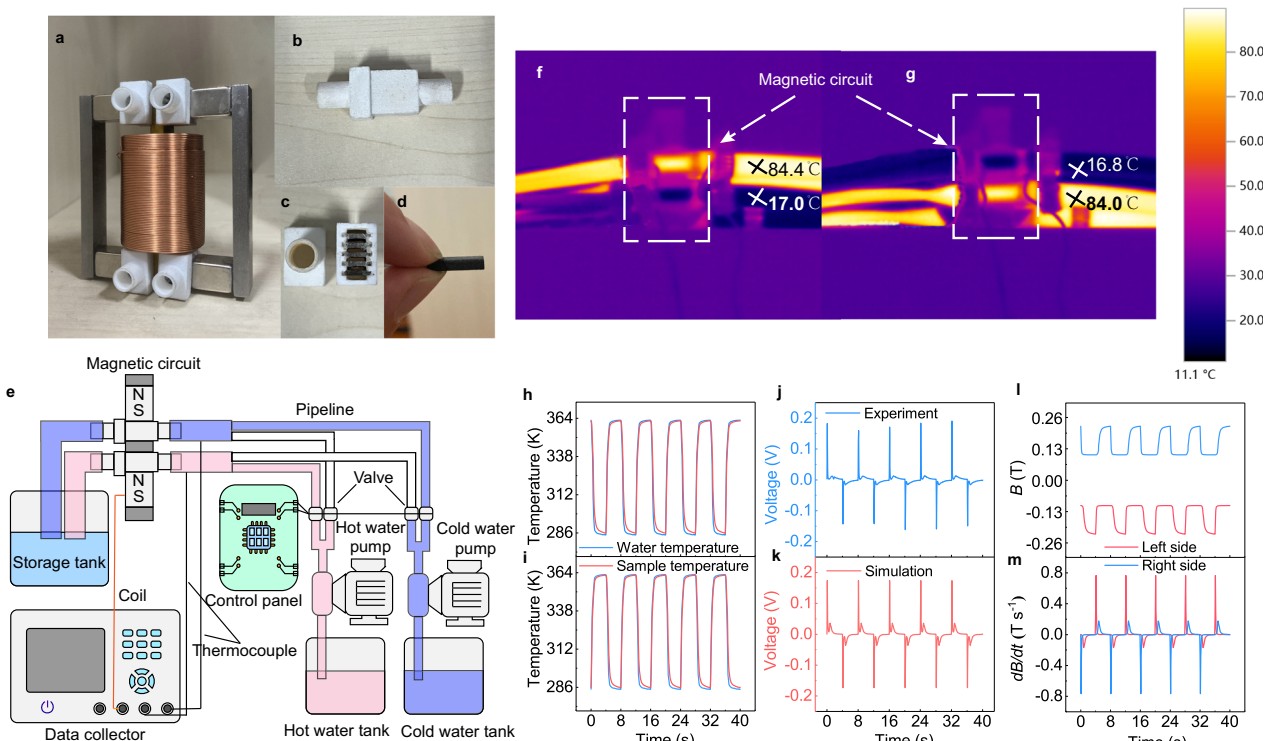

**Fig. 3 | Device construction and performance test. a–d** Assembled magnetic circuit and associated components. **a** Photo of the magnetic circuit. **b** Sample cabin. **c** The MCSs in each sample cabin. **d** The MCS with a size of 6.5 × 12 × 1 mm³. **e** Schematic diagram of the test process of the TMG system. The system consists of a magnetic circuit, a data collector, a control system, cold and hot water pumps, cold and hot water tanks, a recovery tank, pipelines, and valves. Use two K-type thermocouples to connect the data collector to record the temperature change of the water temperature at the left and right sides of the inlet. The coil is connected to the data collector, and the induced voltage generated by the TMG is recorded. **f, g** Infrared image of TMG. **f** Cold water flows through the left (lower) side while hot water flows through the right (upper) side of the magnetic circuit. **g** Hot water flows through the left (lower) side while cold water flows through the right (upper) side of the magnetic circuit. **h–m** TMG performance test. **h** The temperature curve of the water and MCS on the left inlet. **i,** The temperature curve of the water and MCS on the right inlet. **j** The induced voltage curve generated by the TMG. **k** The numerical simulation of the induced voltage. **l** The magnetic flux density **B** and **m,** the corresponding $d\mathbf{B}/dt$ curve on the left and right sides of the magnetic circuit. (Source data are provided as a Source Data file).

opposite directions by alternating hot and cold water. Each heating and cooling cycle is 8 s. The temperature changes fast within the first ~1 s, and then gradually reaches the stable temperature, i.e., 287 K for cold end and 361 K for hot end. This fact indicates the fast thermal response of MCS. The stable temperatures are close to the temperatures obtained by the infrared camera.

Figure 3j shows the induced $V$ generated by the TMG as a function of time. When the left MCS is cooled and the right MCS is heated, a positive $V$ peak of ~0.19 V is observed at 0.1 s. A small peak of ~0.016 V is seen at ~0.8 s. Such successive large and small peaks appear throughout the cycle. The induced $V$ can be also obtained by finite element simulations using COMSOL Multiphysics, as shown in Fig. 3k. The simulated large and small $V$ peaks are 0.17 V and 0.036 V, respectively, which are in a good agreement with the experimental results, confirming the accuracy of the simulation model. The slight difference between experimental and simulated results, especially the lower experimental value of small $V$ peak, might be attributed to the error and imperfect thermal conduction in experiment. Furthermore, the magnetic flux density **B** of the MCS was obtained by the simulation to analyze the $V$ curve. The directions of **B** on both sides are opposite to each other; the **B** on the left side is defined as negative, and the **B** on the right side is defined as positive, as shown in Fig. 3l. When the left MCS is cooled while the right MCS is heated, the absolute value of **B** on the left side increases while the one on the right side decreases. The $T_C = 292$ K of Gd is closer to the cold end temperature of 283 K compared to the hot end temperature of 363 K, hence, the FM-PM magnetic transition would occur earlier during heating than the PM-FM transition during cooling. The MCSs on the two sides do not turn ON/

OFF at the same time even though the hot and cold valves are switched simultaneously. Therefore, the decrease of |**B**| on the right side occurs within 0.1 s, and the increase of |**B**| on the left side occurs later at 0.8 s. Furthermore, it is seen from Fig. 3h,i that the $dT/dt$ at 292 K during heating is always larger than that during cooling, which leads to a faster decrease of |**B**| than the increase of |**B**|, as shown in Fig. 3l. The corresponding $d\mathbf{B}/dt$ is obtained, as shown in Fig. 3m. Due to the asynchronous switching of the left and right MCSs, successive large and small $d\mathbf{B}/dt$ peaks (0.76 T/s and 0.16 T/s) are obtained in each cycle, and the ratio between the large and small $d\mathbf{B}/dt$ peaks is in a good agreement with the ratio of simulated $V$ peaks. Consequently, it results in the successive large and small $V$ peaks according to Faraday's law.

## High performance of TMG

To obtain high TMG performance, we further studied the effect of non-structural parameters on the TMG performance through a combination of experiments and finite element simulations. Figure 4a shows the peak $V$ and average power $P_{avg}$ of the TMG as a function of cold end temperature. The $P_{avg}$ is obtained using the following equation[20]:

$$P_{avg} = \frac{1}{t_{cycle}} \cdot \int_{t}^{t+t_{cycle}} \frac{V(t)^2}{R} \, dt \qquad (2)$$

where $V$ is the induced voltage, $R$ is the resistance of coil (11 Ω), $t$ is the time, and $t_{cycle}$ is the time required for each cycle. When the cold end temperature is 298 K, a temperature which is higher than the $T_C$ of Gd, the magnetic transition cannot occur, and so the MCS would be OFF all the time. Thus, no power is generated by the TMG. As the cold end

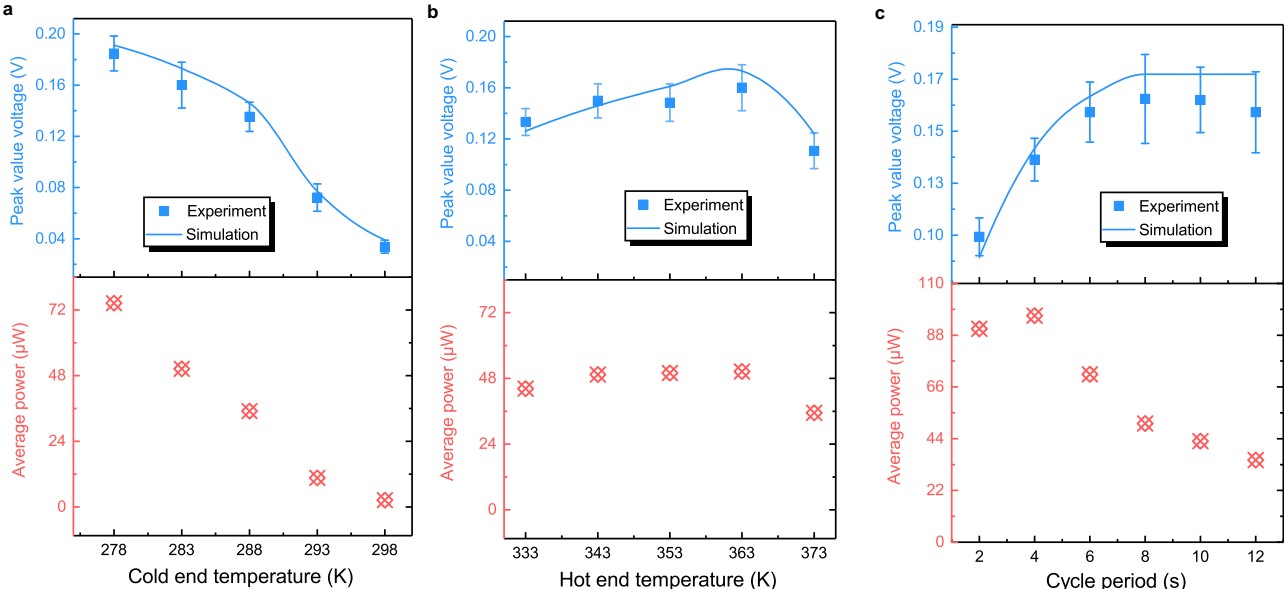

**Fig. 4 | Effect of non-structural parameters on the TMG performance through a combination of experiments and finite element simulations. a** The peak $V$ and average power $P_{avg}$ of the TMG as a function of cold end temperature with a fixed hot end temperature of 363 K. **b** The peak $V$ and $P_{avg}$ as a function of hot end temperature with a fixed cold end temperature of 283 K. **c** The peak $V$ and $P_{avg}$ as a function of cycle period. The error bar is the standard deviation of multiple cyclic voltage peaks during the testing time. (Source data are provided as a Source Data file).

temperature decreases, the PM-FM transition of MCS can be more fully induced by colder water, causing a larger magnetic flux density change $\Delta B$ (Supplementary Fig. 11). The lower cold end temperature leads to larger $dT/dt$, as shown in Supplementary Fig. 11. Larger $\Delta B$ and $dT/dt$ values result in higher $dB/dt$, increasing the $V$ and average power significantly: from 0.03 V and 2.5 μW at 298 K to 0.18 V and 74.5 μW at 278 K.

Similarly, the peak $V$ and $P_{avg}$ as a function of hot end temperature are shown in Fig. 4b. The peak $V$ and $P_{avg}$ also increase gradually with the hot end temperature increasing from 333 K to 363 K, but the increase rate is not as fast as that with the decrease of cold end temperature. Supplementary Fig. 12 reveals that, although $dT/dt$ increases largely with an increase of hot end temperature, the $\Delta B$ does not increase very much. Since the hot end temperature of 333 K is substantially higher than $T_C = 292$ K, the Gd could be converted into the complete PM state by the hot fluid, the $\Delta B$ during the magnetic transition reaches the maximum value. Further increase of hot end temperature does not cause a larger decrease of $B$, e.g., the $\Delta B$ only increase ~0.005 T with the hot end temperature increasing from 333 K to 363 K. Consequently, the increase of $|dB/dt|$ is only 0.16 T/s, much smaller than the increase of $|dB/dt|$ (0.53 T/s) as the cold end temperature decreases from 298 K to 278 K. This fact explains the slow increase of $V$ and $P_{avg}$. When the hot end temperature further rises to 373 K, the hot water pump reaches the highest service temperature, causing the flow rate to decrease from 0.06 L/s to 0.047 L/s. Then, the TMG performance reduces remarkably.

As shown in Fig. 3h, i, the temperature changes very fast at the beginning of each cycle, completing the magnetic transition of MCS within the first few seconds. It suggests that the cycle period could be shortened to obtain more induced power. Figure 4c shows the peak $V$ and $P_{avg}$ as a function of cycle period. The peak $V$ remains stable while the $P_{avg}$ increases significantly with the shortening of cycle period until 6 s. This fact confirms that the cycle period could be reduced to 6 s without affecting the magnetic transition of MCS. Meanwhile, more induced power can be obtained, resulting in the increase of $P_{avg}$. With further reduction of cycle period, the peak $V$ decreases largely, while $P_{avg}$ still increases significantly until 4 s and then reduces slightly at 2 s. The effect of the cycle periods on the TMG performance are described

in detail as shown in Supplementary Figs. 13, 14. The sample temperature curve matches well with the water temperature curve when the cycle period is larger than 6 s. When the cycle period is less than 6 s, a discrepancy between the sample temperature and the water temperature appears. This would result in an incomplete magnetic transition, thus reducing the induced power. Although the induced power is reduced, more power peaks are obtained, and so $P_{avg}$ still increases at 4 s and only slightly reduces at 2 s. This result is in a good agreement with our recent work[20].

**Application potential of the TMG**

These results indicate that the TMG performance is largely influenced by key parameters. Accordingly, we optimized the hot end temperature, cold end temperature, and cycle period to 278 K, 363 K, and 4 s, respectively. Figure 5a compares the peak $V$ and $P_{avg}$ before and after optimization (the parameters before and after optimization are shown in Supplementary Table 2). By optimizing the parameters, we could maintain the induced $V$ at a value as high as 0.16 V, while obtaining more induced $V$ peaks, the $P_{avg}$ is significantly increased from 50.5 μW to 123.3 μW. In our previous study, LaFeSiH/In composites were demonstrated to exhibit excellent TMG performance among typical magnetocaloric effect (MCE) materials. Therefore, we used LaFeSiH/In as MCSs to test the TMG performance. The detailed MCE and TMG performance of LaFeSiH/In MCS are presented in Supplementary Figs. 15, 16. Figure 5b compares the different TMG factors of Gd and LaFeSiH/In materials. It is noted that the TMG performance of LaFeSiH/In is obtained by shifting the working temperature range to ensure that the $T_C$s of both Gd and LaFeSiH/In are located at the same position of their respective working temperature range (Supplementary Fig. 17)[12,30]. The unit peak voltage $V_{unit}$ of 0.034 V/g, max power density $P_{Dmax}$ of 4 mW/cm³, and average power density $P_{Dave}$ of 232 μW/cm³ for LaFeSiH/In are much higher than those of Gd (0.031 V/g, 3.2 mW/cm³, and 156 μW/cm³). Only the conversion efficiency $\eta rel$ of LaFeSiH/In (0.14%) is lower than that of Gd (0.18%). This result confirms that LaFeSiH/In exhibits a better property set. In comparison with the pretzel-like topology which also uses La(Fe, Si)₁₃-based materials[18], our TMG uses nearly 40% less La(Fe, Si)₁₃-based materials but generates a 55% higher $V_{unit}$ and 300% higher $P_{Dmax}$ than the $V_{unit}$ of 0.022 V/g and

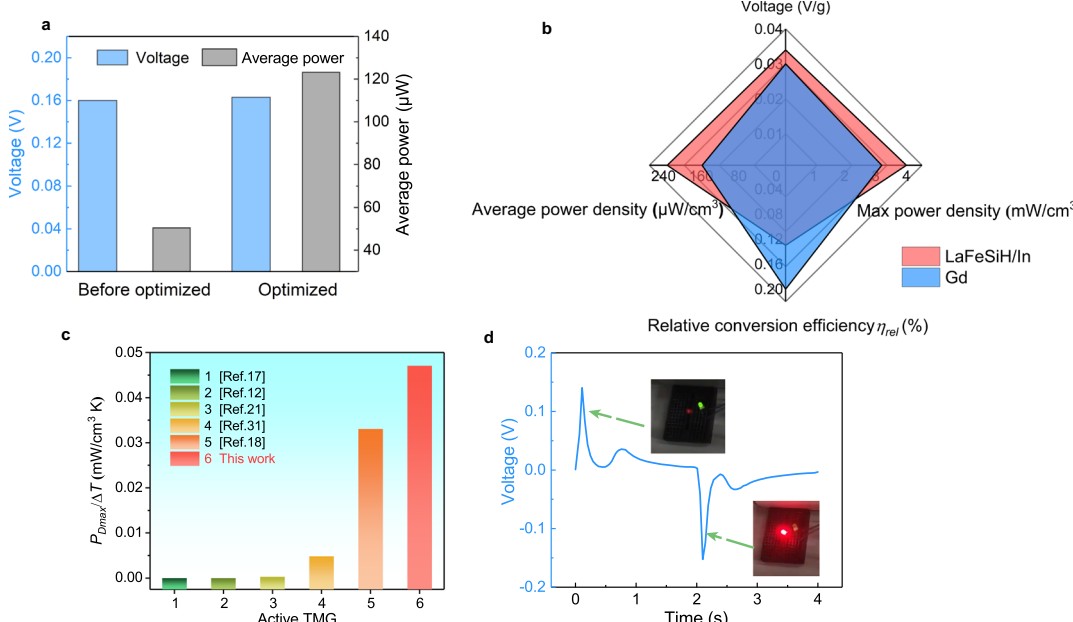

**Fig. 5 | Device performance comparison and application potential demonstration. a** Performance comparison before and after test parameter optimization. **b** Comparison of power generation performance between Gd and LaFeSiH/In.

**c** Comparison of $P_{Dmax}/\Delta T$ of different active TMGs. **d** Utilize devices to light up LEDs. (Source data are provided as a Source Data file).

$P_{Dmax}$ of 0.992 mW/cm³ obtained by pretzel-like topology. Moreover, the $P_{Dmax}$ produced by our TMG is not only much higher than those of other previous TMGs[11,12,16,17,21,22], but also higher than those of TEG[23–26] and PEG[5,27,28]. Furthermore, Fig. 5c compares the $P_{Dmax}/\Delta T$, where $\Delta T = T_{hot}-T_{cold}$, for the present TMG with other active TMGs[12,17,18,21,31]. It is clearly seen that the $P_{Dmax}/\Delta T$ produced by our TMG is much higher than those of other active TMGs, e.g., the $P_{Dmax}/\Delta T$ of our TMG is 0.047 mW/(cm³ K), 42% higher than 0.033 mW/(cm³ K) obtained by pretzel-like topology. Especially, this high $P_{Dmax}/\Delta T$ is even 2 to 3 orders of magnitude higher than those of other active TMGs. Such high TMG performance proves the effectiveness of our topology design of magnetic circuit with MCS.

To demonstrate the practicability of our TMG, we connect it with LEDs. More details about the circuit for lighting up the LEDs are described in Supplementary Note 6. Figure 5d shows that the induced $V$ of our TMG using Gd metal successfully lights up the LEDs. The yellow LED is lit when the TMG generates positive voltage, while the red LED is lit when a negative voltage is generated. Due to the successive big and small $V$ peaks during each heating/cooling process, the LED will flash twice with different brightness. The process of LED lighting is shown in the Supplementary Movie 1. Only a few grams of sample are used in our TMG demo. More power can be induced if more samples are used. Furthermore, we propose a potential utilization scenario of the TMG in future, i.e., multiple groups of TMGs are connected in series, and the $T_C$ of MCS materials in each TMG is in a $T_C$ gradient from one side to the other side. So successive induced power can be obtained as the waste heat goes through these TMGs. This kind of TMG array can convert the waste heat from industry into electric energy for power grid, thus realizing the reutilization of energy.

## Concluding Remarks

In summary, we designed a TMG for low-temperature waste heat recovery. Instead of acting as the coil core or big magnetic yoke in previous TMGs, in our design the magnetocaloric material acts as a small switch which controls the ON/OFF of the magnetic circuit. In comparison with most previous TMGs in which the flux only changes between zero and maximum value, the present topology, like the pretzel-like topology, can also make the magnetic flux in the induction

coil change between the negative and positive maximum values, resulting in an increase of the induced power by a factor of four. Moreover, in comparison with the pretzel-like topology, present design shows simpler design and lower magnetic stray field, and uses less TMM materials but generates higher $V_{unit}$, $P_{Dmax}$, and $P_{Dmax}/\Delta T$ than those generated by pretzel-like TMG. Furthermore, the TMG performance is significantly improved by optimizing the key system parameters through a combination of experiments and finite element simulations. The $P_{Dmax}$ produced by our TMG is not only much higher than those of other active TMGs, but also higher than those of thermoelectric generator and pyroelectric generator. Especially, the $P_{Dmax}/\Delta T$ of our active TMG is even 2 to 3 orders of magnitude higher than those of other active TMGs. Such high TMG performance proves the effectiveness of our topology design of magnetic circuit with MCS. Furthermore, a LED light is successfully lit up by our TMG, and a TMG array is proposed as a potential utilization scenario to harvest the low-grade waste heat.

## Methods
### Material preparation and characterization

All the raw materials were purchased from a commercial vendor with purity above 99.9 wt.%. The Gd metal (>99.95 wt.%) was cut into 20 plates of $6.5 \times 12 \times 1$ mm³ by wire-cutting. The $La_{0.7}Ce_{0.3}Fe_{11.51}Mn_{0.09}Si_{1.4}$ ingot was prepared by induction melting, and then processed into strips by melt-spinning. The strips were subsequently annealed in a high-vacuum quartz tube at 1373 K for 9 days followed by ice-water quenching. Then, The $La(Fe, Si)_{13}$ sample was pulverized into powders with size less than 150 μm, and hydrogenated to saturation in a hydrogen atmosphere of 0.2 MPa at 623 K for 5 days. The $La(Fe, Si)_{13}H_y$ hydride powders were mixed with 30 wt.% pure Indium powder, and then pressed into a $34 \times 20 \times 5$ mm³ mold at 60 MPa and 140 °C for 30 min. After hot pressing, the samples were wire-cut into 20 plates of $6.5 \times 12 \times 1$ mm³ for the MCSs. Their density was measured with the Archimedes method. The magnetization and specific heat were measured using a cryogen-free, cryocooler-based measurement system (VersaLab, Quantum Design Inc.); the thermal transport option of the platform was utilized to determine the thermal conductivity.

## TMG setup

The topology of the TMG magnetic circuit is shown in Fig. 1. The magnetic circuit consists of three yokes (DT4C; $5 \times 12 \times 70$ mm³) which are connected by small permanent magnets ($15 \times 12 \times 10$ mm³) and layered MCSs ($6.5 \times 12 \times 1$ mm³). The permanent magnets with a remanence of ~1.2 T provide a stable magnetic field, and the magnetic flux always circulates through a complete closed loop, which lowers the magnetic stray fields. The center yoke is wound by a copper coil of 1600 turns. The Gd plays as a MCS which controls the ON/OFF of the magnetic circuit. To facilitate heat transfer, the MCS is made into thin plates and layered in the 3D printed sample cabin. Hot and cold water are controlled to flow alternately through the right and left MCSs, thus switching ON/OFF the circuit on each side. K-type thermocouples were used to measure both water and sample temperatures. The induced voltage and temperature were collected by using a data collection system (RIGOL M300). The infrared image during the operation of the TMG is captured by the infrared imager (Testo 890), and then the image is processed by IRsoft published by Testo Inc.

## Numerical Simulations

Finite element simulation was carried out by the COMSOL Multiphysics software (Version 5.4). The heat transfer and magnetic field modules were used to simulate the temperature change of MCS and the induced voltage $V$. The governing equations, boundary conditions, and mesh testing of the numerical model are detailed in Supplementary Note 2. The simulated result was compared with the experimental result to verify the accuracy of the model. Furthermore, the effects of key structural and system parameters on the TMG performance were studied using finite element simulation (Supplementary Note 3, 4).

## Data availability

Source data are provided with this paper. All data needed to evaluate the conclusions in the paper are available within the article and its Supplementary Information files. All raw data generated during the current study are available from the corresponding author upon request. Source data are provided with this paper.

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

## Acknowledgements

H.Z. acknowledges the support by the National Natural Science Foundation of China (Grant No. 52171169) and the National Key Research and Development Program of China (Grant No. 2021YFB3501204). K. M. Q. acknowledges the support by the National Natural Science Foundation of China (Grant No. 52101210). F. X. H. also acknowledges the support by the National Natural Science Foundation of China (Grant Nos. 52088101 and 92263202) and the National Key Research and Development Program of China (Grant No. 2019YFA0704900). K. C. thanks the support by the National Natural Science Foundation of China (Grant No. 52161025).

## Author contributions

H.Z. conceived the project. X.L.L. carried out the main experiments and the finite element simulation. H.D.C. and J.Y.H. measured the TMG property and lit up the LEDs. K.M.Q. and Z.Y.Y. performed the magnetic measurements. L.L.X. prepared the MCS materials. R.V.R., F.X.H., K.C., and Y.L. discussed the results and reviewed the manuscript. X.L.L and H.Z. prepared the manuscript with contributions from all authors.

## Competing interests

The authors declare no competing interests.
