## [Peer Review File · Nature Communications]

High-performance thermomagnetic generator controlled by a magnetocaloric switchEditorial Note: This manuscript has been previously reviewed at another journal that is not operating a transparent peer review scheme. This document only contains reviewer comments and rebuttal letters for versions considered at *Nature Communications*.

REVIEWERS' COMMENTS

Reviewer #4 (Remarks to the Author):

1. The article is well written and the conclusions and claims are well addressed.
2. The methodology is sound and the work is of good quality.
3. The description of the work is sufficiently presented to allow the work to be reproduced.
4. In general, paper is very well written, figures and diagrams are clear, including all the explanations.
4. Following the state of the art work of Brillouin L and Iskenderian HP from 1948, and all the other work done (but not necessarily reported by authors) from that day on in the field of thermomagnetic generators, I unfortunately do not recognize the groundbreaking nature of the work of authors, but rather an incremental improvement of technology.

Reviewer #5 (Remarks to the Author):

The manuscript of Liu et al (High-Performance Thermomagnetic Generator Controlled by a Magnetocaloric Switch) describes the design and testing of a power generator using magnetocaloric material. The design is similar to previous works whereby the principle employed to produce electrical power is by flux change through a coil. A magnetic flux circuit is created using permanent magnets and ferromagnetic flux guides. The preferred flux path is altered by cycling magnetocaloric material through its magnetization-temperature state space. The novelty in the work of Liu et al is in the topographical structure of the flux circuit. The magnetocaloric material is located outside the coil region, thereby providing easy access to effect temperature change through heat transfer with a fluid.

The manuscript is well written and the plots are good (although perhaps too small in some of the multi-panel figures.) The supplemental material provides sufficient information to interpret and replicate the work. While the overall quality of the work is good, there are areas for improvement, particularly around the interpretation of the results and the claims.

The authors state that their design uses magnetocaloric material (MM) as a "switch" (instead of a "yoke" as in other designs) and therefore doesn't require as much material. This is not a general truth. The MM must act as a flux path in order to close the magnetic circuit and efficiently use the flux created by the permanent magnets - hence, it has features of "switch" and "yoke." Furthermore, the claims of needing less MM material are not generally valid. Less material was used by Liu et al than in the Waske device they refer to, however, the amount of MM will constrain the maximum work that can be produced. This is because MM material is fundamentally converting heat to work through variation in the T-s state space. While Liu et al used less material than other devices and achieved increased specific power, it is important to note that absolute power production will scale with amount of material for devices with equivalent efficiency and cycle time. The numerical results indirectly support this as they show how plate thickness impacts performance. Further, the amount of MM will impact the use of flux (i.e. stray field), but also the flux available for work in the MM; an insufficient amount of MM will not maximally utilize the field and there will be "stray" field. Too much MM will result in lower power density. This is an optimization problem for all TMG devices.

The authors claim that "... the magnetocaloric material acts as a switch that controls the magnetic circuit. This makes it not only have all the advantages of the pretzel-like topology such as flux reversal in the induction coil, but also show more advantages than pretzel-like topology, e.g., simpler design, fewer magnetocaloric materials, and lower magnetic stray field." is too general.

Likewise, the phrase "...a colossal TMG performance..." is unjustified. (The word "colossal" seems to have become accepted in describing MM properties, but it has no definition.) The authors report a relative efficiency of 0.14 % for LaFeSiH/In and 0.18 % for Gd (as compared to 0.17 % for the Waske device) - none of these metrics suggest "colossal" performance (nor sufficiently high performance to be commercially viable.) Finally, the closing discussion seems to be out of scope i.e. feeding electricity to the grid. There is a significant amount of work needed to address considerations such as capital cost, efficiency and lifetime.

Overall the work is interesting and useful to other researchers in the field. However, I recommend the authors clarify some of the above points to temper their claims and to improve the paper.

Reviewer #4 (Remarks to the Author):

1. The article is well written and the conclusions and claims are well addressed.
2. The methodology is sound and the work is of good quality.
3. The description of the work is sufficiently presented to allow the work to be reproduced.
4. In general, paper is very well written, figures and diagrams are clear, including all the explanations.
5. Following the state of the art work of Brillouin L and Iskenderian HP from 1948, and all the other work done (but not necessarily reported by authors) from that day on in the field of thermomagnetic generators, I unfortunately do not recognize the groundbreaking nature of the work of authors, but rather an incremental improvement of technology.

Reply: We thank the Reviewer very much for the high recognition of our work.

Reviewer #5 (Remarks to the Author):

The manuscript of Liu et al (High-Performance Thermomagnetic Generator Controlled by a Magnetocaloric Switch) describes the design and testing of a power generator using magnetocaloric material. The design is similar to previous works whereby the principle employed to produce electrical power is by flux change through a coil. A magnetic flux circuit is created using permanent magnets and ferromagnetic flux guides. The preferred flux path is altered by cycling magnetocaloric material through its magnetization-temperature state space. The novelty in the work of Liu et al is in the topographical structure of the flux circuit. The magnetocaloric material is located outside the coil region, thereby providing easy access to effect temperature change through heat transfer with a fluid.

The manuscript is well written and the plots are good (although perhaps too small in some of the multi-panel figures.) The supplemental material provides sufficient

information to interpret and replicate the work. While the overall quality of the work is good, there are areas for improvement, particularly around the interpretation of the results and the claims.

Reply: We greatly appreciate the reviewer's recognition of our work. We have revised our manuscript according to your valuable suggestions. Especially, we enlarged the multi-panel figures as suggested.

The authors state that their design uses magnetocaloric material (MM) as a "switch" (instead of a "yoke" as in other designs) and therefore doesn't require as much material. This is not a general truth. The MM must act as a flux path in order to close the magnetic circuit and efficiently use the flux created by the permanent magnets - hence, it has features of "switch" and "yoke." Furthermore, the claims of needing less MM material are not generally valid. Less material was used by Liu et al than in the Waske device they refer to, however, the amount of MM will constrain the maximum work that can be produced. This is because MM material is fundamentally converting heat to work through variation in the T-s state space. While Liu et al used less material than other devices and achieved increased specific power, it is important to note that absolute power production will scale with amount of material for devices with equivalent efficiency and cycle time. The numerical results indirectly support this as they show how plate thickness impacts performance. Further, the amount of MM will impact the use of flux (i.e. stray field), but also the flux available for work in the MM; an insufficient amount of MM will not maximally utilize the field and there will be "stray" field. Too much MM will result in lower power density. This is an optimization problem for all TMG devices.

Reply: Thank you for your valuable suggestion. We totally agree with the Reviewer that it is inappropriate to simply claim that less magnetocaloric materials (MM) are needed by using it as a "switch" instead of a "yoke" as in other designs. In comparison with the traditional TMG, in which the MM plays as the magnetic core and requires larger amount in order to achieve higher flux change, the MM is located outside the coil region in our present design, and this design makes us use less material than other

devices and achieved increased specific power. However, as pointed out by the Reviewer, our experiments have also verified that the change of the thickness and length of MM plate would affect the magnetic flux and then influence the performance of thermomagnetic generator. Our results confirm that the optimization of the amount and size of MM is important to obtain the best TMG performance.

Based on the suggestion of the Reviewer, we have revised the corresponding expression and no longer emphasize the reduction of MM amount by optimizing the device design. Instead, we only state that our device uses less MM than previous devices and exhibits better TMG performance.

The related revision has been highlighted by red color in Abstract and Conclusion.

The authors claim that "... the magnetocaloric material acts as a switch that controls the magnetic circuit. This makes it not only have all the advantages of the pretzel-like topology such as flux reversal in the induction coil, but also show more advantages than pretzel-like topology, e.g., simpler design, fewer magnetocaloric materials, and lower magnetic stray field." is too general. Likewise, the phrase "...a colossal TMG performance..." is unjustified. (The word "colossal" seems to have become accepted in describing MM properties, but it has no definition.) The authors report a relative efficiency of 0.14 % for LaFeSiH/In and 0.18 % for Gd (as compared to 0.17 % for the Waske device) - none of these metrics suggest "colossal" performance (nor sufficiently high performance to be commercially viable.) Finally, the closing discussion seems to be out of scope i.e. feeding electricity to the grid. There is a significant amount of work needed to address considerations such as capital cost, efficiency and lifetime.

Reply: We thank you very much for this constructive suggestion. "... the magnetocaloric material acts as a switch that controls the magnetic circuit. This makes it not only have all the advantages of the pretzel-like topology such as flux reversal in the induction coil, but also show more advantages than pretzel-like topology, e.g., simpler design, fewer magnetocaloric materials, and lower magnetic stray field." is too general. According to the reviewer's suggestion, we have made corresponding revision in the manuscript (highlighted by red color on pages 2, 4, and 15).

We agree with the reviewer that the present efficiency is not high enough to be called “colossal”, so we have replaced the “colossal” with “high” in the revised manuscript.

Finally, as suggested by the Reviewer, we deleted Figure 5e and modified the closing discussion on page 14.

Overall the work is interesting and useful to other researchers in the field. However, I recommend the authors clarify some of the above points to temper their claims and to improve the paper.

Reply: Thank you for your positive comments. As suggested, the above points have been carefully revised to temper our claims and to improve the paper.

The changes of the content are tracked and highlighted by red color in the revised manuscript.

Finally, we really appreciate that the editor and reviewers treated our manuscript very patiently and gave us these valuable comments. We believe that we have answered all the questions of editor and reviewers. Therefore, we are resubmitting the revised manuscript for your consideration.

Sincerely yours

Hu Zhang

School of Materials Science and Engineering

University of Science and Technology Beijing

Beijing 100083

P. R. China